# A model for drug transport across two membranes of Gram-negative bacteria by an MFS tripartite assembly

Zhaojun Zhong[1,8], Tuerxunjiang Maimaiti [1,8], Matthew L. Jackson[2,3,8], Rui Dong[4,8], Xueyan Gao [1,8], Qing Ouyang[1], Wenqian Wang[1], Jinliang Guo [1], Shangrong Li [1], Wenyu Shang [1], Huajun Liu [1], Hongnian Jiang[4,5], Shuo Zhang[4,5], Ulrich Zachariae [6] ✉, Ben F. Luisi [2] ✉, Yanjie Chao [4,5] ✉ & Dijun Du [7] ✉

Transport of proteins and small molecules across cellular membrane is crucial for bacterial interaction with the environment and survival against antibiotics. In Gram-negative bacteria that possess two layers of membranes, specialized macromolecular machines are required to transport substrates across the cell envelope, often via an indirect stepwise process. The major facilitator superfamily (MFS)-type tripartite efflux pumps use proton electrochemical gradient to extrude drugs in diverse bacterial species, but the architecture of the assembly and structural mechanisms remain elusive. A representative MFS-type tripartite efflux pump, EmrAB-TolC, mediates resistance to multiple antimicrobial drugs through proton-coupled EmrB, a member of the DHA2 transporter family. Here, we report the high-resolution (3.13 Å) structure of the EmrAB-TolC pump, revealing a distinct, asymmetric architecture emerging from the assembly of TolC:EmrA:EmrB with a ratio of 3:6:1 and contacts that are essential for the pump assembly. Key residues involved in drug transport are identified and corroborated by mutagenesis and antibiotic sensitivity assays. The structural and functional data support a model for one-step drug transport by the MFS pump across the entire envelope of Gram-negative bacteria.

Antimicrobial resistance of Gram-negative bacteria is one of the most devastating issues in modern healthcare systems worldwide. The complex architecture of the cell envelope in Gram-negative bacteria presents a formidable obstacle for the penetration and transport of antimicrobial drugs. As one way to develop drug resistance, these organisms have evolved an array of specialized macromolecular nanomachines capable of extruding internalized drugs back to the environments[1,2], for example the tripartite efflux pumps, double membrane-spanning nanomachines that are closely related with type I secretion systems (T1SSs)[1,2]. To cross the double membranes of Gram-

[1]School of Life Science and Technology, ShanghaiTech University, Shanghai, China. [2]Department of Biochemistry, University of Cambridge, Cambridge, UK. [3]MRC Toxicology Unit, University of Cambridge, Cambridge, UK. [4]Shanghai Institute of Immunity and Infection, Chinese Academy of Sciences, Shanghai, China. [5]Shanghai Institute of Materia Medica, Chinese Academy of Sciences, Shanghai, China. [6]Biological Chemistry and Drug Discovery, Faculty of Life Sciences, University of Dundee, Dundee, UK. [7]State Key Laboratory of Metabolic Dysregulation & Prevention and Treatment of Esophageal Cancer, Tianjian Laboratory of Advanced Biomedical Sciences, School of Convergence Medicine, The Fifth Affiliated Hospital of Zhengzhou University, Zhengzhou University, Zhengzhou, Henan, China. [8]These authors contributed equally: Zhaojun Zhong, Tuerxunjiang Maimaiti, Matthew L. Jackson, Rui Dong, Xueyan Gao. ✉e-mail: u.zachariae@dundee.ac.uk; bfl20@cam.ac.uk; yjchao@ips.ac.cn; dudijun7025@gmail.com

negative bacteria, substrates may be exported directly from the cytosol to the outer membrane (OM) in a one-step manner, or indirectly in a two-step fashion in which substrates are first moved to the periplasm through a transporter system on the inner membrane (IM), and then translocated across the OM by another[2]. Of three classes of tripartite efflux pumps, two operate on the two-step transport mechanism[3], including the resistance-nodulation-cell division (RND) types exemplified by AcrAB-TolC[4,5] and the ATP-binding cassette (ABC) types represented by MacAB-TolC[6]. The MFS-type tripartite efflux pumps are believed to employ a one-step transport mechanism like T1SSs; however, the molecular details of the one-step process remain unclear[7,8].

The MFS-type tripartite efflux pumps are conserved in many Gram-negative bacterial pathogens[7,8]. The prototype MFS-type EmrAB-TolC pump of *Escherichia coli* plays an essential role in mediating multi-drug resistance and contributes to virulence phenotypes[9–12]. This pump consists of three components: a TolC OM channel, an EmrB IM transporter, and an EmrA periplasmic adapter which bridges the two transmembrane proteins[10]. Understanding of the overall structure and organization of the EmrAB-TolC pump is currently limited to the structural data of individual, separate components of its homologs[7,8]. The *Aquifex aeolicus* EmrA structure exhibits a linear arrangement of α-helical hairpin, lipoyl, and β-barrel domains. A conserved loop in the β-barrel domain is much longer than that of other adapters, which is disordered in the determined structure[13]. EmrB belongs to the DHA2 transporter family and, unlike the well-characterized DHA1 family for which structures in various conformations are available[14–17], the DHA2 family has been less well characterized. However, recent studies on an analogous DHA2 transporter QacA has revealed critical acidic residues essential for substrate recognition and transport[18]. Structural analysis of the DHA2 transporter MHAS2168 from *Mycobacterium hassiacum* revealed the extension of two transmembrane (TM) helices, TM11 and TM12, into the periplasm, which might interact with a lipoprotein for substrate translocation[19]. The stoichiometry of the EmrAB-TolC assembly remains undefined, with studies reporting conflicting models of either 3:6:1 or 3:6:2 TolC:EmrA:EmrB ratios[13,20,21]. The high-resolution structure of the full pump assembly and the mechanism of the pump are currently unknown.

In this study, we present the cryo-EM structure of the *E. coli* EmrAB-TolC assembly, revealing how the pump confers multi-drug resistance to antibiotics. The pump structure pinpoints several key residues that are required for drug export, mutation of which abrogate bacterial resistance to multiple antibiotics. The architecture of the EmrAB-TolC pump supports a one-step drug transport model that directly transport antimicrobial drugs across the entire cell envelope of Gram-negative bacteria, offering additional insights on the mechanisms of antimicrobial resistance.

## Results

### Engineering stable and functional EmrAB-TolC complexes

Yousefian et al. demonstrated that the native tripartite EmrAB-TolC complex can be obtained, but the yield is insufficient for cryo-EM analysis[10], because the protein complex is highly susceptible to both dissociation and precipitation. To stabilize the pump assembly, we first modified EmrB by fusing a thermostabilized apocytochrome b562RIL (BRIL) protein and an ALFA tag to the N- and C-termini of the protein[22–24], respectively. We then engineered an EmrB-EmrA fusion protein with a flexible linker that connected the modified EmrB and EmrA. We observed that the N-terminus of EmrA and the C-terminus of EmrB are likely adjacent to the cytoplasmic side, and EmrA possesses a single N-terminal transmembrane helix. Consequently, EmrA was fused to the C-terminus of modified EmrB via a flexible poly-glycine-serine linker, thereby maintaining the correct membrane topology of the pump components (Supplementary Fig. 1a).

Co-expression of the fused EmrB-EmrA protein, free EmrA, and TolC led to the formation of a complex (engineered pump-FA).

However, despite successful expression, the complex tended to dissociate or precipitate during purification, resulting in protein concentrations of only ~0.1 mg/mL. Therefore, the N-terminal segment (residues 1−47) of free EmrA (containing the N-terminal TM α-helix) was replaced by the AcrA signal sequence (residues 1−27, containing residue C25) to direct periplasmic localization. Residues 1−24 of the AcrA signal sequence were subsequently cleaved, leaving C25 available for modification by a palmitoyl acid chain, which anchored EmrA to the inner membrane (IM) (Supplementary Fig. 1a)[25,26]. The modified free EmrA, EmrB-EmrA fusion protein and TolC were successfully expressed and copurified, and the complex (engineered pump-EA) exhibited modest stability during purification, achieving a concentration of ~0.7 mg/ml.

The genomic *emrAB* is expressed poorly in *E. coli* under laboratory growth conditions, and its knockout does not measurably affect resistance[27]. However, mutations that increase the expression of the pump lead to increased resistance[11]. Therefore, we induced the expression of the pump to carry out antibiotic sensitivity assays, which revealed that the wild-type EmrAB-TolC pump conferred drug resistance to nitroxoline, nalidixic acid and CCCP (Supplementary Fig. 1b, c). The engineered EmrAB-TolC constructs, pump-EA and pump-FA, were also expressed and evaluated through bacterial antibiotic sensitivity assays. The results revealed that expression of pump-FA conferred resistance to nitroxoline, nalidixic acid, and CCCP, thereby confirming the functional activity of the tripartite efflux complex in vivo (Supplementary Fig. 1c). In contrast, the EmrAB-TolC pump-EA exhibited no resistance activity, suggesting that the N-terminal α-helix of EmrA is indispensable for pump function (Supplementary Fig. 1c).

### Quaternary structure of the EmrAB-TolC pump

The structure of EmrAB-TolC was solved using single-particle cryo-EM. Four maps were reconstructed: two maps for pump-EA achieved overall resolutions of 3.13 Å and 3.14 Å, respectively; and two maps for pump-FA reached 3.59 Å and 3.66 Å resolution, respectively (Supplementary Figs. 2a–g and 3a–g). Local resolution examination of these maps indicated a resolution ranging between 3.0 Å to 7.0 Å (Supplementary Figs. 2h,I,3h,i). The map quality of EmrAB-TolC pump-FA was insufficient for reliably constructing the full pump structure, whereas the maps for pump-EA were of sufficient quality to allow building models for residues 47−390 of EmrA and residues 13−498 of EmrB (Fig. 1; Supplementary Fig. 2j; 4a; 5a, c). Nevertheless, the model derived from the EmrAB-TolC pump-EA map fits well within the EmrAB-TolC pump-FA map (Supplementary Fig. 3j; Supplementary Fig. 4b; Supplementary Fig. b, d; Supplementary Fig. 6). The models built from both maps show minimal differences, indicating that the pump-EA variant does not disrupt the assembly or structure of the complex. The map for pump-FA was used to construct models for the extra residues 17−46 of EmrA (Supplementary Figs. 6; 7).

The pump has an elongated shape consisting of a TolC trimer, an EmrA hexamer and an EmrB monomer (Fig. 1; Supplementary Fig. 6). In the case of pump-FA, the model of the N-terminal TM helix of the EmrA protomer 4 spans the entire IM, while the partial models of the other five EmrA protomers are shorter (Supplementary Figs. 6; 7a–c).

The positions of EmrB and TolC delineate the IM and OM boundaries, respectively, with the long axis of the pump assembly extending ~320 Å into the periplasm (Fig. 1; Supplementary Fig. 6). This is comparable to the dimensions observed for other types of tripartite assemblies, such as AcrAB-TolC and MacAB-TolC[4–6].

### The EmrA hexamer forms a nanochannel in the pump assembly

The cryo-EM map revealed five sequentially ordered structural units in EmrA: the singular N-terminal α-helix and the α-helical hairpin, lipoyl, and β-barrel domains, as well as a C-terminal α-helix (Supplementary Fig. 7a–c). The N-terminal α-helix is observed for one of the six EmrA protomers, and this helix makes a small contact surface with the EmrB

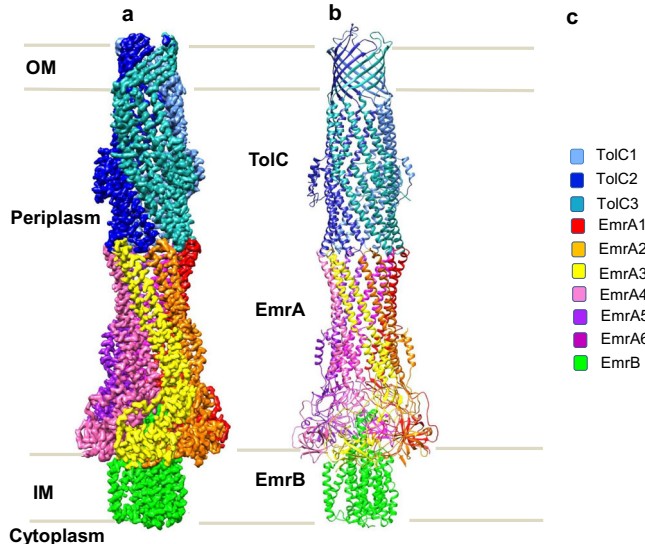

**Fig. 1 | Overall structure of the EmrAB-TolC efflux pump. a** CryoEM map of pump-EA (Map-1). Subunits are color-coded as in **c**. **b** Ribbon representation of the pump with the color code shown in **c**. The TolC trimer spans the outer membrane, with its helical end extending into the periplasm, where it engages the EmrA hexamer. The β-barrel domains of EmrA interact with the periplasmic portion of EmrB.

near the inner leaflet of the inner membrane (Supplementary Fig. 6a, b). The N-terminal TM helix anchors the periplasmic units to the IM (Supplementary Fig.7b). Adapter proteins, such as AcrA and MacA, utilize a MP (membrane proximal) domain for interaction with the periplasmic extension of IM transporters[4–6]. EmrA and its homologs, as well as the adapter proteins of T1SS, however, lack such a MP domain (Supplementary Figs. 8 and 9). The α-helical hairpin domain structure of *E. coli* EmrA shares structural similarity with that of *A. aeolicus* EmrA, despite being shorter in length (Supplementary Figs. 8a, b)[13]. The structures of both the lipoyl and β-barrel domains are similar to those of other adapters (Supplementary Figs. 8 and 9), despite their sequences being less conserved (Supplementary Fig. 10). A notable difference is the presence of a loop (termed the conserved loop of β-barrel domain, β-CL), comprising residues 294–317, in the β-barrel domain of EmrA (Supplementary Figs. 7b; 11a), which is conserved among EmrA homologs (Supplementary Fig. 12a). A long coil follows the C-terminus of the β-barrel domain, folding back to engage with the side of the adjacent lipoyl domain in the pump assembly. This coil precedes an α-helix at the terminus (residues 375–390) that, together with the lower portion of the α-helical hairpin, forms a three-stranded coiled-coil, mimicking the α-helical hairpin domain of CusB (Supplementary Fig. 7b)[28].

The α-helical hairpin, lipoyl and β-barrel domains of EmrA assemble into a hexameric structure reminiscent of AcrA and MacA within the MacAB-TolC and AcrAB-TolC pumps (Supplementary Fig. 7c; 9)[4,6]. The helical hairpin domains create an α-helical barrel nanochannel. The lipoyl domains form a ring encompassing an internal chamber (lipoyl chamber) that extends the nanochannel. The analogous hexameric assembly of MacA presents a gating ring formed by conserved glutamine residues in the lipoyl domain[6], but this is notably absent in the EmrA assembly. The six β-CL loops within the β-barrel domains (β-CL1 to β-CL6) vary in conformation, creating a structural unit, which forms the wall of a cavity (β-barrel cavity) (Supplementary Fig. 11a–c). Residues L302 and L303 in the β-CL hexamer form a hydrophobic ring-1 on the IM distal side of β-barrel cavity, providing a narrow entryway to the lipoyl chamber. Conversely, residues I313, V315 and V316 generate a hydrophobic ring, ring 2, on the IM proximal side of β-barrel cavity, potentially situated on the surface of the inner membrane (Supplementary Figs. 11b, c; 12a). The β-CL1 contains an

alternative route (termed β-CL1PH) linking the lipoyl chamber and β-barrel cavity (Supplementary Fig. 11c).

## A monomer of EmrB containing a periplasmic helix bundle mediates pump assembly

EmrB, belonging to the DHA2 family, comprises 14 transmembrane helices, of which TM1−TM12 adopt the characteristic fold of MFS transporters such as MdfA and EmrD[15,16]. The six-helix bundles, formed by TM1−TM6 (termed N-domain) and TM7−TM12 (termed C-domain), create a V-shaped transporter (Fig. 2a−c) with a pseudo-twofold symmetry axis perpendicular to the membrane plane. Two more TM helices, $TM_A$ and $TM_B$, are inserted into the cytoplasmic loop bridging the N-domain and C-domain, but against the periphery of the protein core, similar to YbgH and PepTso[29–31]. $TM_A$ and $TM_B$ counterparts of QacA and Tet(L) are known to be important for their functions[18,32,33].

Structural analysis of EmrB revealed a central aqueous cavity located in the core of the membrane that opens toward the periplasmic space, indicating that EmrB adopts an outward-open state (Supplementary Fig. 13a−c). Helices TM4 and TM5 in the N-domain, along with TM10 and TM11 in the C-domain, constitute a cytoplasmic gate. Residues I120 and P121 (TM4), V147 (TM5), F379 (TM10), N403 and R406 (TM11) interact to seal the base of the cavity (Supplementary Fig. 13c). Within the central cavity, we identified a proton-titratable residue D29, located in motif D, which we hypothesized would be critical in the antiporter function of EmrB[34]. The corresponding residues of D34 in the TM1 of QacA and MfdA have previously been shown to be crucial for substrate recognition[16,17]. In our structure, D29 forms a salt bridge with R109 and hydrogen bond with Q112, respectively (Fig. 2d). All three of these residues are highly conserved among EmrB homologs (Supplementary Fig. 12b). The $EmrB_{D29N}$ or $EmrB_{R109A}$ mutation was found to abrogate bacterial resistance to nalidixic acid, nitroxoline and CCCP, supporting the hypothesis that these residues are involved in proton translocation during transport (Fig. 2e). Interestingly, the absence of either EmrA or EmrB abolishes efflux pump activity (Fig. 2e).

Everted membrane vesicle assay was employed to directly evaluate the substrate/H[+] antiport activity of EmrB alone, the EmrAB complex, and the full EmrAB-TolC efflux pump, incorporating both wild-type and mutant forms of EmrB. Proton transport was monitored in everted membrane vesicles using the pH-sensitive fluorophore 9-amino-6-chloro-2-methoxyacridine (ACMA). An inward proton gradient (acidic interior) was established via lactate oxidation and detected as the quenching of ACMA fluorescence. Subsequent addition of the substrates nalidixic acid (Nal) or 2-chlorophenylhydrazine hydrochloride (CHH) elicited fluorescence dequenching, indicative of an outward proton flux consistent with H + /Nal or H + /CHH antiport activity. This substrate-dependent dequenching was significantly more pronounced in vesicles expressing the wild-type EmrAB-TolC complex than in empty-vector controls. The minimal residual response observed in control vesicles is likely attributable to endogenous *E. coli* transport activity for these substrates.

Notably, vesicles expressing either EmrB alone or the EmrAB subassembly (lacking TolC) exhibited no Nal- or CHH-induced dequenching above background levels, demonstrating that antiport activity was entirely abrogated in these systems. Surprisingly, the antiport activity of the full EmrAB-TolC pump was unaffected by charge-neutralizing substitutions at the conserved EmrB carboxylate residues D29 and D221. Vesicles expressing the D29N/D221N mutants exhibited dequenching levels equivalent to those of the wild-type EmrAB-TolC pump. The vesicle data suggest that proton antiport can still occur when D29 and D221 are neutralized, implying that proton coupling may involve a more distributed network of residues (Supplementary Fig. 14).

To examine the potential functional relevance of protonating the D29 carboxylate group, we performed atomistic molecular dynamics

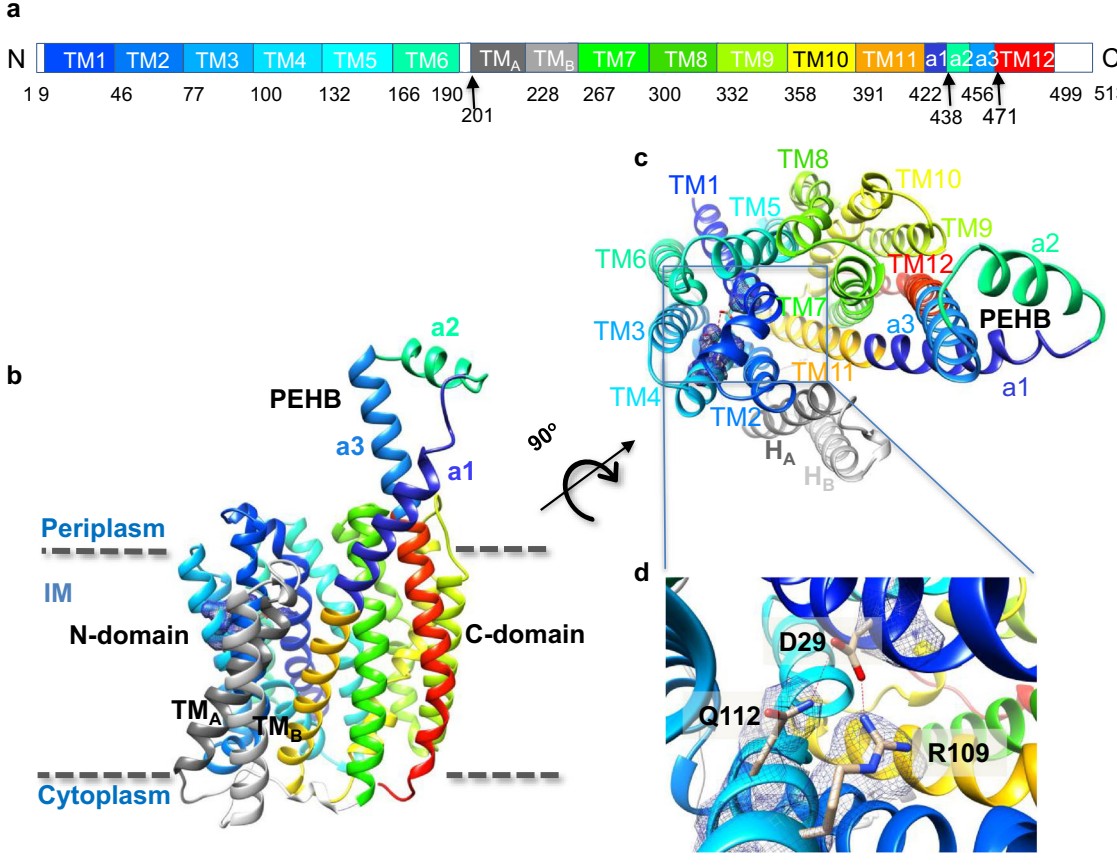

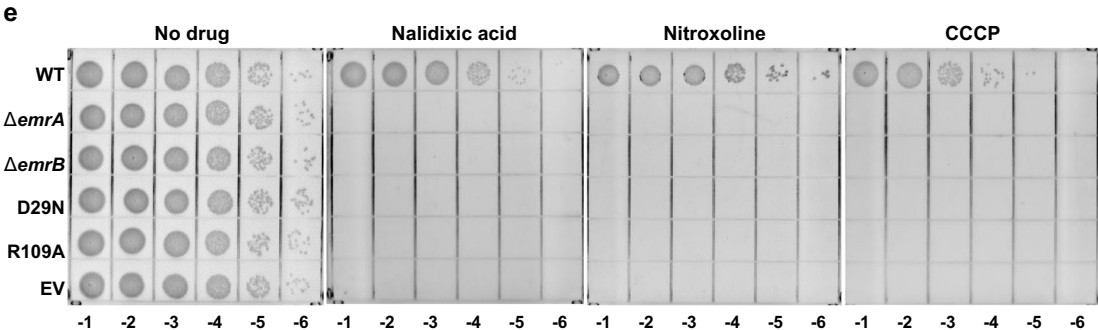

**Fig. 2 | Structure and function of EmrB. a** Linear representation of EmrB (the color code in **a** is also used for the domains and subdomains in **b**). The numbers below the color-coded bar indicate the approximate positions within the amino acid sequence of EmrB, corresponding to transmembrane helices TM1–TM12 in the membrane, as well as their adjacent helices and loops located outside the membrane. **b, c** Ribbon representations of EmrB viewed parallel to the membrane and perpendicular to the membrane on the periplasmic side, respectively. **d**, Organization of critical residues potentially involved in H⁺ translocation and coupling. Cryo-EM densities corresponding to residues R109 and Q112 were well resolved, whereas the density for D29 was less well-defined, likely due to radiation-induced damage from electron exposure. Red dashed lines designate a salt bridge between D29 and R109 and a hydrogen bond between D29 and Q112. **e** Mutants of EmrA/EmrB exhibit loss of resistance to nalidixic acid, nitroxoline and CCCP. Antibiotic sensitivity assays of *E. coli* C43(DE3) *ΔacrAB* expressing wild-type EmrAB-TolC, EmrA-TolC (*ΔemrB*), EmrB-TolC (*ΔemrA*), EmrAB_{D29N}-TolC, or EmrAB_{R109A}-TolC, were carried out as in Supplementary Fig. 1a. "WT" and "EV" denote wild-type EmrAB-TolC and the empty vector containing the negative control, respectively.

(MD) simulations of EmrB embedded into mixed POPE/POPG (3:1) lipid bilayers, comparing the dynamics of EmrB with charged or neutral D29 in triplicate simulations of 1 μs length each. In all cases, the protonation state change of D29 from charged to neutral led to disruption of the salt bridge between D29 and R109 (Supplementary Fig. 15a, b). At the same time, the H-bond between D29 and Q112 was broken. Within a few ns, the protonated D29 side chain reorientated to form a tight H-bond with the I170 backbone carbonyl group in all simulations, while the slightly weaker interaction with N129 was retained (Supplementary Fig. 15b, c).

To detect medium-to-long range effects of the D29 protonation state change, we used our mutual-information based analysis method SSI[35] to identify conformational changes in EmrB that were correlated specifically with this change. Supplementary Fig. 16 shows that the conformations of neighboring residues on helix 1 were strongly affected by the reorientation of D29 (e.g., Q26, I32). Furthermore, residues on helix 6 near I170 displayed conformational transitions linked to the rearrangement (F171, I173). At a greater distance from D29, but proximal to the R109 side chain, we found residues S57 and I106 to be impacted by D29 protonation. Excluding the conformational

dynamics of loop regions due to their raised flexibility, the longest-distance conformational change observed to be correlated with the protonation change involved residue R406, located in the putative alternating-access binding pocket of EmrB. Taken together, these findings demonstrate that protonating the D29 side chain affects the EmrB conformation in regions both proximal and distal to D29.

Parts of the 13th and 14th helices (TM11 and TM12) of EmrB protrude into the periplasm and form a structural unit named periplasmic extended helix bundle (PEHB) (Fig. 2b, c). The PEHB is composed of an N-terminal helix (a1) following TM11, a C-terminal helix (a3) preceding TM12, and a horizontal helix (a2) parallel to the membrane, along with the loops connecting these helices. The most pronounced difference between EmrB and other MFS transporters is the distinct PEHB structure, which is key for pump assembly (Fig. 2b, c; Supplementary Fig. 17a–d).

## Comparison between EmrB, QacA and MHAS2168

The determined structures of EmrB, QacA (PDB ID 7Y58) and MHAS2168 (PDB ID 8PNL) give an insight into the architecture and functional variety of the DHA2 family transporters (Supplementary Fig. 17a–c)[19,33]. Despite a shared evolutionary origin, these transporters display distinct structural configurations and mechanisms adapted to their specific roles. All three proteins have the characteristic MFS fold: an N-terminal and a C-terminal domain collectively contributing 12 core transmembrane helices, with two additional transmembrane helices inserted between these domains (for a total of 14). Each structure is shown in outward-facing state and differentiated by a notable periplasmic extension (PEHB) between helices TM11 and TM12. In EmrB, the PEHB has an N-terminal helix (α1) at the end of TM11, a C-terminal helix (α3) right before TM12, and a horizontal helix (α2) oriented parallel to the membrane (Supplementary Fig. 17a). The PEHB is critical for assembling the functional tripartite efflux pump. The corresponding PEHB of QacA (denoted as extracellular loop 7, EL7) is in a form of a helix-turn-helix, and the interaction between EL7 and the first extracellular loop (EL1) is fundamental for efficient antibacterial efflux activity (Supplementary Fig. 17b). The PEHB of MHAS2168 presents yet another variant: TM12 itself extends into the periplasm as a four-turn α-helix, connected by a linker loop to a ~2.5-turn extension of TM11, a structural arrangement proposed to facilitate the transfer of triacylglycerides (TAG) to the lipid-binding protein LprG (Supplementary Fig. 17c). These structural diversities are particularly related to functional specialization. EmrB and QacA construct a negatively charged substrate-binding cavity that is more likely to accommodate cationic antimicrobial compounds, whereas the vestibule of MHAS2168 is highly hydrophobic, consistent with its role in lipid (TAG) transport. EmrB contains a single conserved acidic residue (D29), which is proposed to function as a proton translocation site in its drug/$H^+$ antiport mechanism. In contrast, MHAS2168 possesses two candidate proton translocation residues (D35 and E157). QacA exhibits multiple acidic residues distributed within its vestibule, generating a region of intense negative electrostatic potential. Molecular dynamics (MD) simulations identify D411 as a putative proton translocation site in QacA. Although D34 in QacA aligns with the functionally critical residues D29 (EmrB) and D35 (MHAS2168) and is indispensable for QacA activity across all substrate classes, no experimental evidence implicates D34 in proton translocation.

## Interfacial contacts between EmrA and EmrB

Interactions of EmrB with other components of the tripartite assembly appear to be required for efflux activity, suggesting that the interactions allosterically modulate the activity of the MFS protein. The β-barrel domain of EmrA mediates interaction with EmrB, with the β-CL region of EmrA playing a key role through interactions with the PEHB of EmrB (Fig. 3a; Supplementary Fig. 18a, b). The a2 in PEHB penetrates the hydrophobic ring-1 in the β-CL region of EmrA (Fig. 3a, b; Supplementary Fig. 18a). Since a2 is larger than the narrow entrance

formed by hydrophobic ring-1, it becomes trapped above the hydrophobic ring-1, effectively blocking the narrow entrance (Fig. 3b; Supplementary Fig. 11b, c; Supplementary Fig. 18a). The EmrA$_{L302Q/L303Q}$ mutation abrogated pump-mediated resistance to nalidixic acid, nitroxoline and CCCP, indicating that the hydrophobic ring-1 plays an essential role in pump assembly (Fig. 3d, e; Supplementary Fig. 18a). Below the hydrophobic ring-1, helices a1 and a2 in the PEHB tilt towards one side of the β-barrel cavity, making contacts only with β-CL1 to β-CL3 (Fig. 3b; Supplementary Fig. 18a, b).

In addition, the cytoplasmic surface of the EmrA β-barrel domains directly engages with the periplasmic region of EmrB. The protruding loops in EmrB, positioned between helices H1−H2, H7−H8 and H9−H10, fit snugly into the concave pockets in the β-CL region of EmrA. The N-terminal TM helices of EmrA insert into the membrane, stabilizing the interaction interface. The N-terminal α-helix 4 of EmrA directly contacts TM$_B$ of EmrB, which may affect the function of the pump (Supplementary Fig. 19). Helices a1 and a3 of PEHB, along with other periplasmic regions between the transmembrane helices of EmrB, position along the side of the β-barrel cavity. The internal β-barrel cavity formed is sufficiently large to accommodate all known EmrB substrates. The cavity in EmrB opens to adjacent β-barrel cavity in EmrA, allowing the substrate to be released from the former into the latter (Fig. 3c). Given the blockage of the narrow entrance by a2, the route of β-CL1PH in β-CL1 serves as the sole link between the lipoyl chamber and the β-barrel cavity (Fig. 3b, c).

The hydrophobic ring-2 in β-CL region of EmrA assists in the formation of a sealed continuous channel between the outward-open cavity (Cavity$_{out}$) in the EmrB and the β-barrel cavity by resting on the surface of the cellular membrane (Supplementary Fig. 11b, c). The EmrA$_{I313Q/V315T/V316T}$ mutation in hydrophobic ring-2 abrogated resistance to nalidixic acid, nitroxoline and CCCP, indicating that the ring is critical for the pump assembly (Fig. 3d, e; Supplementary Fig. 11c).

## Prediction of EmrB in its inward-open conformation

EmrB has been suggested to adopt an inward-open state for substrate recognition and binding during transport, thereby forming a periplasmic gate[36]. However, in both our cryo-EM structure and all 5 models predicted with high-confidence scores by AlphaFold3, EmrB is consistently displayed in an outward-open state (Fig. 4a; Supplementary Fig. 20a, b)[37]. In an attempt to predict EmrB in its inward-open state, we utilized the multimer feature of AlphaFold3 to generate a prediction of EmrB in complex with six molecules of EmrA. In all 5 of the models generated by AlphaFold3, we were able to observe EmrB in a distinct conformation (Fig. 4b; Supplementary Fig. 20c, d). Based on the disappearance of the outward-open cavity and the presence of a new and enlarged cytosolic cavity, we hypothesized that AlphaFold3 had also predicted the inward-open state of EmrB that has not yet been observed experimentally, and which is consistent with our mutagenesis data and with known MFS transport mechanisms (Fig. 4c; Supplementary Fig. 13d, e)[36]. Using a homology model of EmrB in the inward-open state in conjunction with these predictions, we identified several residues which we propose would be important in the process of translocation. Residues N62, V147, I148 and I288 are all situated along the wall of the cytosolic-facing cavity (Supplementary Fig. 13e). The corresponding residues in MdfA are involved in drug recognition (Supplementary Fig. 17d–f)[17]. In support of AlphaFold3, mutations of N62, V147, and I288 indeed reduced pump-mediated resistance to nitroxoline and nalidixic acid. I148 was crucial for resistance to CCCP, whereas it appeared dispensable for nitroxoline and nalidixic acid (Fig. 4f, g).

Molecular ensemble docking of the substrates nalidixic acid and nitroxoline shows that the likely main drug binding site of EmrB is located in the vicinity of D29, where the substrate molecules are wedged between the side chains of Q26 and V27 (Supplementary Fig. 17g, h).

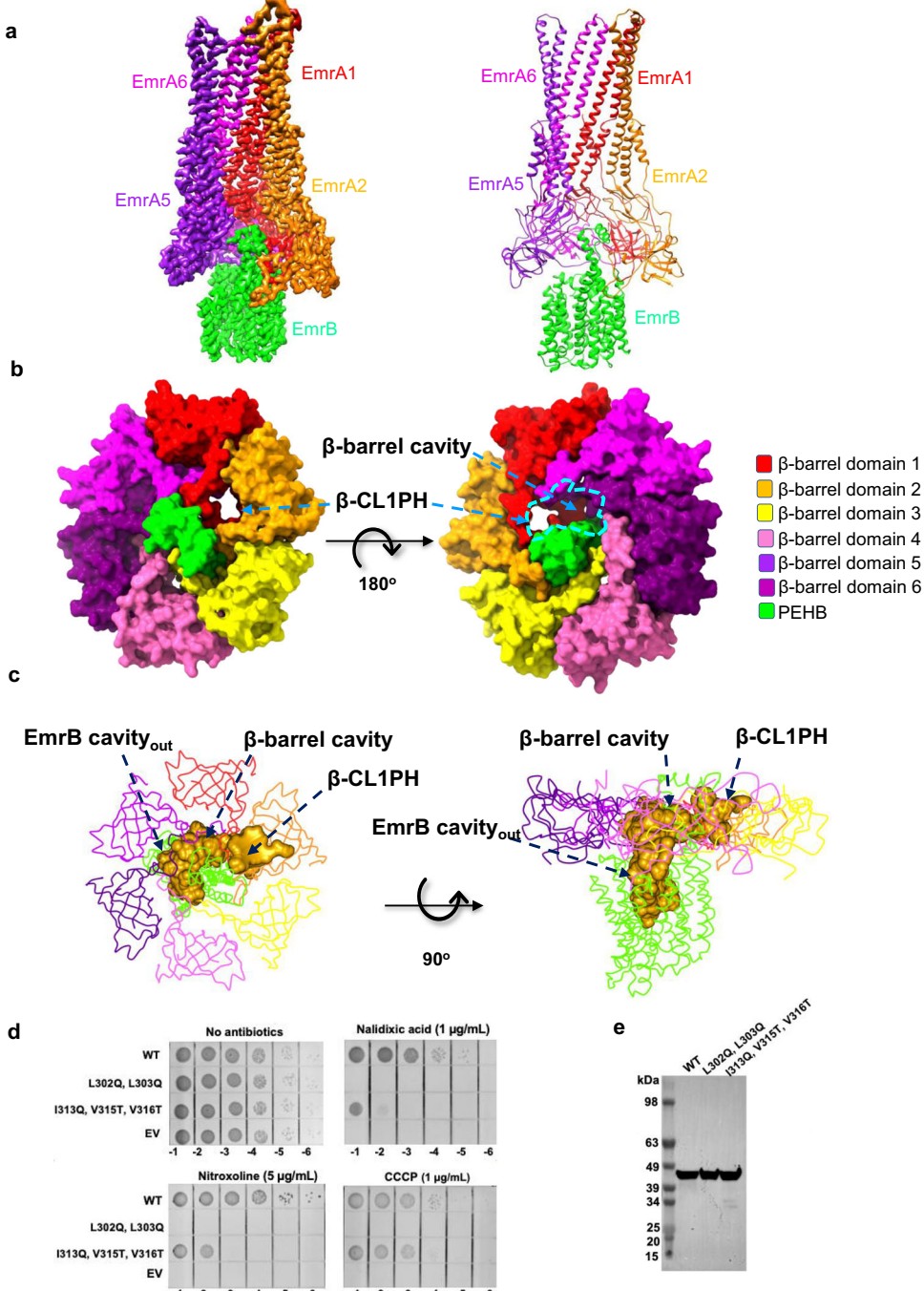

**Fig. 3 | Interactions between EmrA and EmrB. a** Map and model showing the interfacial contacts between EmrA and EmrB viewed parallel to the membrane. Protomers 3 and 4 of EmrA are omitted for clarity. **b** Surface representations of the β-barrel domains of EmrA and the PEHB (periplasmic helix bundle) subdomain of EmrB, showing contacts between those domains, as well as the internal β-barrel cavity and β-CL1PH (viewed perpendicular to the membrane at the inner membrane distal (left) and proximal (middle) sides, respectively). Domains and subdomains are color-coded accordingly (right). **c** The transport pathway in the EmrB and EmrA β-barrel domains, viewed perpendicular (left) and parallel (right) to the membrane. Domains are represented as α-traces and are color-coded as in **b**. The outward-open cavity of EmrB, the β-barrel cavity, and the β-CL1PH pathway are shown in gold and

the path β-CL1PH are colored red and were calculated using Caver 3.0[44]. **d** Mutants of EmrA exhibit loss of resistance to nalidixic acid, nitroxoline, and CCCP. Antibiotic sensitivity assays of *E. coli* C43(DE3) *ΔacrAB* expressing wild-type EmrAB-TolC, EmrA$_{L302Q/303Q}$-EmrB-TolC, or EmrA$_{I313Q/V315T/V316T}$-EmrB-TolC, were carried out as in Supplementary Fig. 1c. The assays were performed three times independently. "WT" and "EV" denote wild-type EmrAB-TolC and the empty vector containing the negative control, respectively. **e** Western blot assays demonstrate that the expression levels of EmrA mutants in the EmrAB-TolC pumps are similar to those of the wild type ones, and confirm that these pump components are properly localized in the membrane.

Altogether, these results support the predicted model for the inward-open state and demonstrates the importance of specific residues within EmrB in exhibiting poly-specific drug recognition during the transport process.

## Proposed structural transitions of EmrB

Upon close inspection of the structures of EmrB in different states, we detected changes in the chemical environments of residues already revealed to be important in our mutagenesis studies. To provide

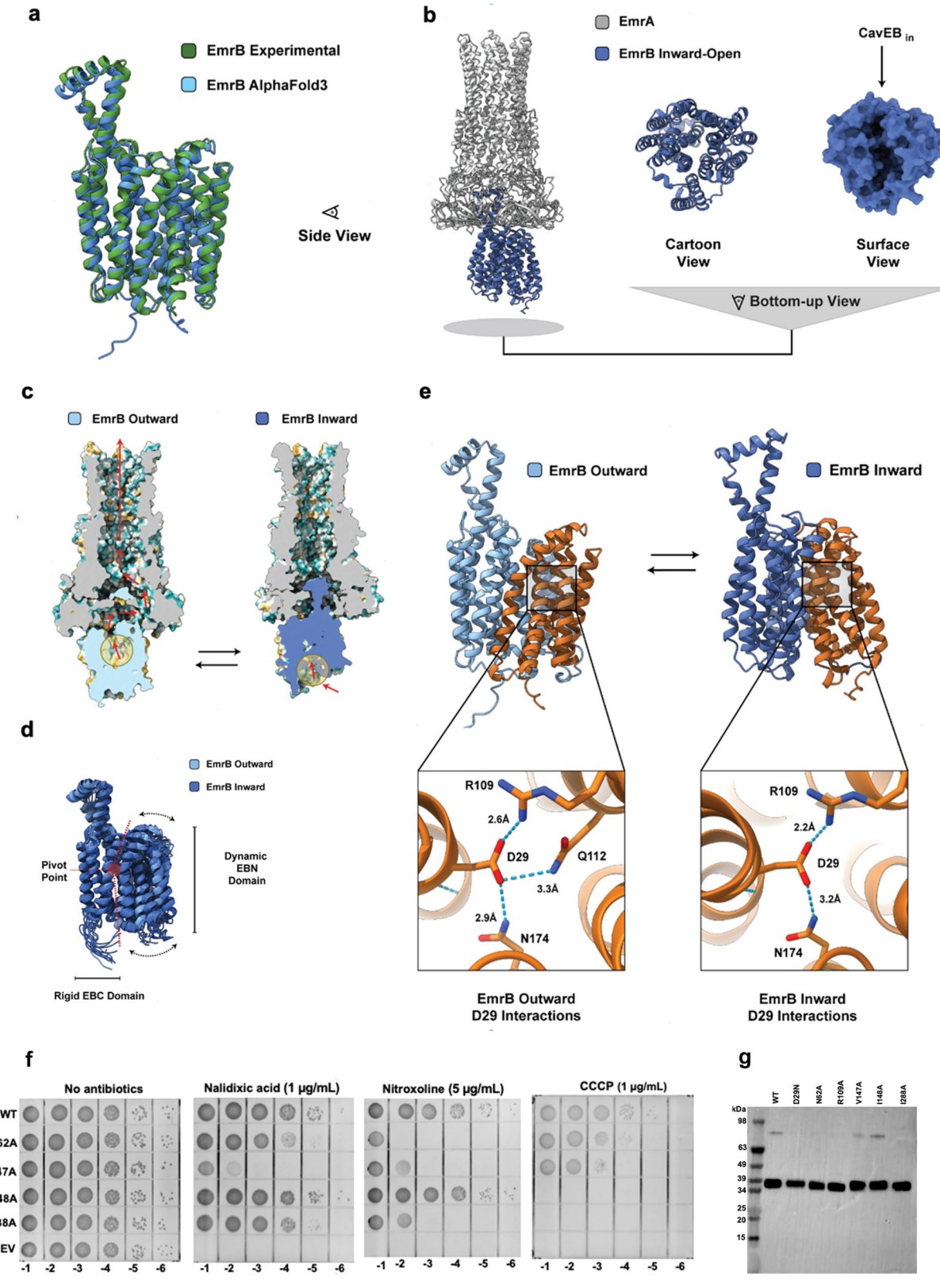

further evidence that AlphaFold3 had indeed predicted the inward-open state and to try gain insight into conformational changes within EmrB, we employed a modified version of AlphaFold2, known as AF-Cluster, to obtain predictions of the various states which EmrB could adopt[38]. By grouping the multiple-sequence alignments based on sequence similarity within the MFS family and subsequently performing AlphaFold2 predictions, we predicted a plethora of structures

representative of the transition between the outward-open to inward-open state of EmrB (Fig. 4e). Interestingly, 75% of these high-confidence predictions could be classified into structures representative of the inward-open states, providing further evidence of this conformation being adopted by EmrB. Furthermore, these predictions also provided insight into the conformational dynamics that EmrB is likely to undergo in shuttling substrates, analogous to the transitions

**Fig. 4 | Conformational changes of EmrAB. a** Alignment of the experimentally determined structure of EmrB with the top-rank prediction of EmrB using Alpha-Fold3. **b** AlphaFold3 and AF-Cluster predict EmrB in its inward-open conformation. This conformation has a large cytosolic binding cleft. **c** Snapshots of the hydrophobic inner section of EmrAB highlight differential cavities involved in the transport of antibiotics. **d** AlphaFold2 structures with high confidence scores were predicted from discrete sequence clusters and overlaid using ChimeraX. These structures reveal the conformational dynamics of EmrB as it transitions between an outward-open to inward-open state through changes in the positions of helices H1-H6 about a central pivot point. **e** D29 exhibits 3 polar contacts in its outward-open state, in comparison to just 2 contacts in the inward-open state. **f** Mutants of EmrB exhibit loss of resistance to nalidixic acid, nitroxoline and CCCP. Antibiotic sensitivity assays of *E. coli* C43(DE3) Δ*acrAB* expressing wild-type EmrAB-TolC, EmrAB$_{N62A}$-TolC, EmrAB$_{V147A}$-TolC, EmrAB$_{V148A}$-TolC, or EmrAB$_{I288A}$-TolC, were carried out as in Supplementary Fig. 1a. The assays were performed three times independently. **g** Western blot assays demonstrate that the expression levels of EmrB mutants in the EmrAB-TolC pump are similar to those of the wild type and confirm that these pump components are properly localized in the membrane.

proposed for other members of the MFS family[36]. The helices within the C-domain appear to remain static, whereas helices TM1 – TM6 in the N-domain are suspected to be highly dynamic. The helices appear to pivot at a central point, in support of a 'rocker-switch' mechanism of transport for EmrB, common to other MFS antiporters[39]. Altogether, these results provide evidence of an inward-open conformation of EmrB, as well as insights into the structural transitions of EmrB that are important in the energy-dependent flux of antibiotics by EmrAB-TolC. The interactions of EmrA-EmrB change in the structural transitions of EmrB, and may provide the path of allosteric communication of these components.

### Interactions between EmrA and TolC

A short helix-turn-helix motif located within the α-helical hairpin domain of EmrA mediates tip-to-tip interactions with the periplasmic end of TolC, similar to the interactions observed between the homologous MacA and TolC (Supplementary Fig. 21a–c)[6]. Engagement between TolC and EmrA directly opens the periplasmic aperture of TolC, creating a continuous nanochannel with a width of 25–30 Å. TolC remains open during the pumping process, as observed previously for RND-type AcrAB-TolC and MexAB-OprM pumps[4,6,40].

## Discussion

Antibiotic resistance of pathogenic bacteria is a growing clinical problem. Drug efflux pumps in the bacterial cell envelope play important roles in multidrug resistance. In the current study, the *E. coli* EmrAB-TolC pump was shown to confer resistance to several different antimicrobial drugs, including nalidixic acid, nitroxoline and CCCP (Supplementary Fig. 1c). The EmrAB-TolC structure identifies key residues that contribute to such multi-drug resistance (Figs. 3d; 4f). The conserved hydrophobic ring-1 of EmrA is crucial for the assembly and functioning of the pump and is accessible to the external environment via the open nanochannel in the full pump assembly (Supplementary Figs. 12a; 22), thus, it represents a promising target for the development of drugs that inhibit this class of pump.

Integrating structural data with MD simulations, we propose a drug transport model of the EmrAB-TolC pump. The structure of the EmrAB-TolC tripartite pump represents the resting apo state structure. The combined data with the Alphafold3 prediction structures are consistent with a one-step transport pathway that bypasses the periplasm (Fig. 5). EmrA and EmrB possess β-CL and the PEHB structure units, respectively (Fig. 2b; Supplementary Fig. 11b, c). The six β-CL loops vary in conformation, one of which develops a path (β-CL1PH) linking the lipoyl chamber and the β-barrel cavity (Fig. 3b, c; Supplementary Fig. 11a–c). Residues L302 and L303 in the β-CL hexamer form a hydrophobic ring-1, onto which the PEHB anchors (Fig. 3; Supplementary Fig. 11b, c; Supplementary Fig. 18a). EmrA lacks the MP domain, which enables EmrA and EmrB to form a secure conduit and its hydrophobic ring-2 strengthens this conduit (Fig. 3d; Supplementary Fig. 11b, c). The structure of the EmrAB-TolC pump delineates the complete transport pathway from the cavity$_{out}$ of EmrB to the cell exterior (Fig. 3c; Supplementary Fig. 22). As a homolog of QacA and MdfA[16,17,41], EmrB in the IM could adopt an inward-open state to facilitate substrate recognition and binding from the inner leaflet of the IM

and cytoplasm, as supported by AlphaFold3 predictions, homology modeling and mutagenesis studies (Figs. 4f; 5; Supplementary Fig. 13d, e). Subsequently, the substrate is transported to the cavity$_{out}$ via an alternating access mechanism, characteristic of MFS transporters (Fig. 5)[36]. The protonation of D29 transforms EmrB to an inward-open state again. The periplasmic regions between the transmembrane helices of EmrB shift toward the central axis, which partially occupy the space in the β-barrel cavity and cause it to shrink. The compaction of β-barrel cavity drives substrates toward the lipoyl chamber and the nanochannels formed by EmrA and TolC. This β-barrel cavity contraction has been proposed in the AlphaFold3-predicted EmrAB structure (Fig. 4c). The transport pathway involves several essential elements: the inward- and outward-open cavities in EmrB, the β-barrel cavity, β-CL1PH, the lipoyl chamber, and the nanochannels formed by EmrA and TolC (Supplementary Fig. 13a–e; Supplementary Fig. 22). These components form a secure conduit for substrate release to the cell exterior (Fig. 5). The gating helices in EmrB allow substrates to flow against concentration gradients and prevent backflow during operation (Supplementary Fig. 13a–e), resulting in the export of drugs across the IM and OM to the extracellular milieu.

At present, six families of efflux pumps are known to be involved in drug transport: primary transporters of the ATP-binding cassette (ABC) family, and secondary transporters including the major facilitator (MFS), resistance/nodulation/cell division (RND), small multidrug resistance (SMR), multidrug and toxic compounds extrusion (MATE), and proteobacterial antimicrobial compound efflux (PACE) families. In Gram-negative bacteria, all these drug transporters are located in the IM. The members of five families of drug transporters (MFS, SMR, MATE, PACE and ABC) usually function as independent units to translocate substrates across the IM bilayer[3]. Drugs on the periplasmic side must be further translocated across the OM. Structures of RND-type AcrAB-TolC and ABC-type MacAB-TolC reveal that the IM components AcrB and MacB collect substrates from the outer leaflet of the IM and periplasm, and efflux them to the extracellular milieu through the tripartite assembly (Supplementary Fig. 23a, b)[4–6]. It is likely that the five transporter systems cooperate with the double membrane-spanning RND-type or ABC-type tripartite pumps to deliver substrates across the OM to the cell exterior. Thus, these tripartite efflux pumps employ a two-step mechanism of periplasmic substrate capture and transport (Supplementary Fig. 23a, b)[2,3]. On the other hand, MFS-type tripartite pumps appear to utilize a one-step mechanism to transport drugs across the entire cell envelope, bypassing the periplasm (Supplementary Fig. 23c). The closely related T1SS assemblies, such as HlyDB-TolC, are believed to employ similar processes for the secretion of protein substrates[1]. The HlyDB-TolC pump could form a tripartite assembly. Similar to EmrA, the adapter protein HlyD lacks an MP domain (Supplementary Fig. 8)[8,42], which may enable HlyD and HlyB to form a secure conduit. The ABC family transporter HlyB could drive the secretion of substrate HlyA across double membranes of the cells via an alternating access mechanism. We note that RND-type tripartite pumps can employ a one-step transport mechanism across the outer membrane for certain substrates, such as beta-lactam antibiotics[43], but it is rather different mechanistically from the EmrAB-TolC pump.

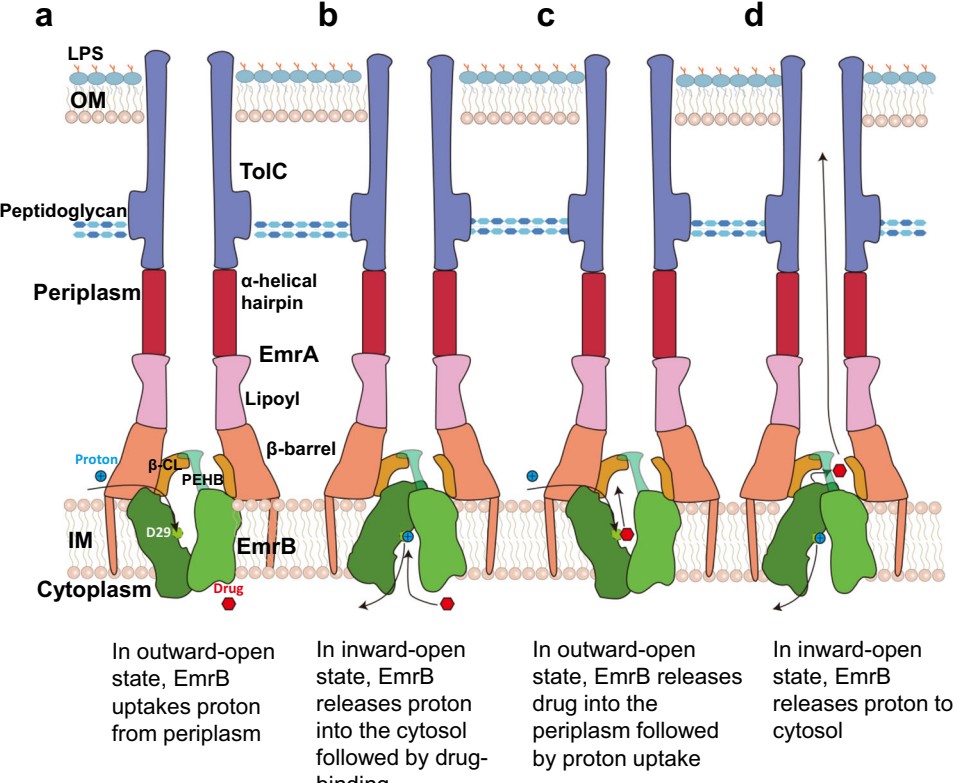

**a** In outward-open state, EmrB uptakes proton from periplasm

**b** In inward-open state, EmrB releases proton into the cytosol followed by drug-binding

**c** In outward-open state, EmrB releases drug into the periplasm followed by proton uptake

**d** In inward-open state, EmrB releases proton to cytosol

**Fig. 5 | Schematic cartoon diagrams of the one-step drug transport mechanism of EmrAB-TolC. a** The resting apo tripartite pump with EmrB in the outward-open state. Helix a2 in the PEHB (periplasmic helix bundle) of EmrB anchors onto a hydrophobic ring-1 in the β-CL region of EmrA to facilitate pump assembly. Engagement between TolC and EmrA directly opens the periplasmic aperture of TolC. Exposure to a low pH periplasmic space leads to protonation of the acidic residue D29 of EmrB. **b** The protonation of D29 changes its salt bridge and hydrogen bond network with residues R109 and Q112, transforms the conformation of EmrB to an inward-open state, allowing H⁺ release from D29 to the cytosol, and

drug binding to a recognition site. **c** Drug binding to EmrB triggers a structural transition that exposes the binding pocket to the periplasm. The drug is then released into the β-barrel cavity. **d** The protonation of D29 returns EmrB to an inward-open state again. As a result, the periplasmic regions between the transmembrane helices of EmrB shift toward the central axis, partially filling the space in the β-barrel cavity. The contraction of the β-barrel cavity propels substrates along the β-CL1PH, the lipoyl chamber toward the nanochannel formed by EmrA and TolC.

The EmrAB-TolC, AcrAB-TolC and MacAB-TolC pumps all utilize the TolC OM channel, and have structurally similar periplasmic adapter proteins that form hexameric channels, but their IM components are derived from distinct transporter families. The periplasmic adapter proteins have evolved to complement the specific IM transporters. As a result, each type of tripartite pump utilizes a distinct pump assembly and gating mechanism that contributes to its function. The ability of these proteins to recognize and transport various substrates largely depends on the specific domains within their IM transporter components, and interactions of the components are proposed to allosterically modulate the activity of the inner membrane component. Here, we observe extensive interactions between EmrA and EmrB that change with conformational switch. These interactions are proposed to modulate the activity of EmrB so that it does not function as a stand-alone transporter, as occurs for example in the well-studied MFS proteins QacA and MfdA. Understanding the similarities and differences between tripartite multidrug efflux pumps and the basis for their allostery is critical for developing strategies to combat multidrug resistance, a global health challenge.

## Methods
### Construction of EmrAB-TolC complex overexpression vectors
The *BRIL* gene was synthesized by AZENTA (Suzhou, China) and amplified using the brilpet-F and bril-link-R primers. The *emrA*, *emrB*, *tolC*, and *acrA* genes were amplified from genomic DNA of *E. coli* K-12

strain MG1655. The *emrB* (37–1494 bp) gene fragment was amplified using primers emrB-link-F and emrBGS-R. The full-length *emrA* gene was first amplified using primers emrAGS-F and emrApet-R; the pETDuet-1 plasmid was linearized using primer pair Petduet_F and Petduet_R; and the *BRIL*, *emrB*, and *emrA* DNA fragments were inserted into the pETDuet-1 vector using NovoRec plus One step PCR Cloning Kit (Novorprotein, Shanghai, China), resulting in pETDuet-*BRIL-emrB-polyGlySer-emrA-6His*. The *ALFA-tag* sequence was subsequently inserted into the 3′ end of *emrB* by site-directed mutagenesis, generating pETDuet-*BRIL-emrB-ALFA-polyGlySer-emrA-6His* (abbreviated as pET_BA).

Using primers TolCinf_F and TolCFLAGXhoI_R, the *tolC* gene was amplified, and the product served as a template for a second amplification using primers TolCinf_F and TolCFLAG_inf_R. The resulting *tolC-FLAG* DNA fragment was inserted into pRSFDuet-1 (digested with *Nde*I) using the in-fusion ligation method of the NovoRec plus One step PCR Cloning Kit (Novorprotein, Shanghai, China), thereby generating pRSFDuet-*tolC-FLAG*. The *emrA* DNA fragment was amplified using primers emrAinsert_F and emrAinsert_R and inserted into the pRSFDuet-*tolC-FLAG* vector using the in-fusion cloning method (Novorprotein, Shanghai, China), resulting in pRSFDuet-*emrA-tolC-FLAG* (abbreviated as pRSF_AC).

Primer pair ΔemrAinsert_F and emrAinsert_R was used to amplify the truncated *ΔemrA* gene (142–1173 bp), and the *ΔemrA* DNA fragment was inserted into the pRSFDuet-*tolC-FLAG* vector using the In-Fusion cloning method (Novorprotien, Shanghai, China), generating

pRSFDuet-*ΔemrA-tolC-FLAG*. The N-terminal 81 bp region of *acrA* (*acrAsignal*) was amplified using the acrAs_F and acrAs_R primer pair, and the resulting *acrAsignal* DNA fragment was inserted into the 5' end of *ΔemrA* in pRSFDuet-*ΔemrA-tolC-FLAG* using the In-fusion cloning method (Novorprotien, Shanghai, China), generating pRSFDuet-*acrAsignal-ΔemrA-tolC-FLAG* (abbreviated as pRSF_ΔAC).

The *emrAB* genes were amplified from genomic DNA of *E. coli* K12 strain MG1655 using primers EmrAB_F and EmrAB_R, the pETDuet-1 plasmid was amplified using primers Petduet_V_F and Petduet_V_R, and the *emrAB* fragment was inserted into pETDuet-1 using an In-Fusion Ligation Kit, resulting in pETDuet-*emrAB*.

The site-directed mutants EmrAB$_{D29N}$ and EmrAB$_{R109A}$ were produced using pETDuet-*emrAB* as a template with the primer pairs emrB$_{D29N}$_F/emrB$_{D29N}$_R and emrB$_{R109A}$_F/emrB$_{R109A}$_R, resulting in pETDuet-*emrAB$_{D29N}$* and pETDuet-*emrAB$_{R109A}$*.

The site-directed mutants EmrAB$_{N62A}$, EmrAB$_{V147A}$, EmrAB$_{V148A}$, and EmrAB$_{I288A}$ were created by using pETDuet-*emrAB* as a template with the primer pairs emrB$_{N62A}$_F/emrB$_{N62A}$_R, emrB$_{V147A}$_F/emrB$_{V147A}$_R, emrB$_{V148A}$_F/emrB$_{V148A}$_R, and emrB$_{I288A}$_F/emrB$_{I288A}$_R, resulting in pETDuet-*emrAB$_{N62A}$*, pETDuet-*emrAB$_{V147A}$*, pETDuet-*emrAB$_{V148A}$*, and pETDuet-*emrAB$_{I288A}$*.

The site-directed mutants EmrA$_{L302Q/L303Q}$B and EmrA$_{I313Q/V315T/V316T}$B were produced using pETDuet-*emrAB* as a template with the primer pairs emrA$_{L302Q/L303Q}$_F/emrA$_{L302Q/L303Q}$_R and emrA$_{I313Q/V315T/V316T}$_F/ emrA$_{I313Q/V315T/V316T}$_R, resulting in plasmids pETDuet-*emrA$_{L302Q/L303Q}$B* and pETDuet-*emrA$_{I313Q/V315T/V316T}$B*.

The enzyme used to introduce the mutations above-mentioned was purchased from CloneAmp HiFi PCR Premix (Clontech, Germany). Then the modified plasmids above-mentioned were amplified and extracted using SPARKeasy Mini Plasmid Ultra-Fast Kit (Shandong Sparkjade Biotechnology Co.,Ltd.).

Primers used to generate all constructs are listed in Supplementary Table 1.

## Overexpression and purification of *E. coli* EmrAB-TolC

The plasmids pET_BA and pRSF_ΔAC were used to co-express and purify EmrAB-TolC pump-EA, which contains the EmrB-EmrA fusion protein, TolC and a modified free EmrA with the N-terminal residues 1–47 replaced by residues 1–27 of AcrA. The 6xHis-Tag at the C-terminus of the EmrB-EmrA fusion protein was initially used to pull down the co-expressed pump components. Next, the FLAG-Tag at the C-terminus of TolC was used to isolate the fully assembled EmrAB-TolC pump. BRIL and ALFA-Tag was used to stabilize EmrB in the complex. The *E. coli* C43 (DE3) *ΔacrAB* strain was transformed with plasmids pET_BA and pRSF_ΔAC. From an agar plate containing appropriate antibiotics, a single colony was picked and inoculated into 20 mL of LB medium supplemented with 50 μg mL$^{-1}$ kanamycin and 100 μg mL$^{-1}$ carbenicillin in a 50 mL centrifuge tube. The culture was incubated at 37 °C with shaking at 220 rpm for 4–5 h.

A 10 mL culture aliquot was then transferred to 1 L of 2×YT medium containing antibiotics in a 2 L baffled flask. The same incubation conditions were applied. Cells were cultured until a density (absorbance at 600 nm, A$_{600}$) of 0.5–0.6, induced with 0.1 mM isopropyl-β-D-thiogalactoside (IPTG), the temperature was reduced to 20 °C, and culturing was continued overnight.

The cells were harvested by centrifugation, and the cell pellet from 6 L of culture was resuspended in 200 mL of lysis buffer containing 20 mM Tris (pH 8.0) and 300 mM NaCl. Afterwards, 2 mL of 100×EDTA-Free Protease Inhibitor Cocktail (APExBIO, USA) was added to a final concentration of 1×. Lysozyme and DNase I were added to final concentrations of 5 mg/mL and 5 U/mL, respectively. The cell mixture was incubated at 4 °C for 1 h before high-pressure homogenization via three passages at 15,000 psi at 4 °C. The lysate was centrifuged to remove cell debris, and the supernatant was ultracentrifuged to pellet the cellular membranes.

Lysis buffer was used to resuspend the membrane pellet. EDTA-free protease inhibitor cocktail (100×) was added to a final concentration of 1×, *n*-dodecyl-β-D-maltoside (DDM) was added to a final concentration of 1.5%, and the mixture was gently stirred at 4 °C for 3 h. The mixture was ultracentrifuged. Imidazole was added to the supernatant to a final concentration of 20 mM, and the mixture was applied to a 1 mL HiTrap Chelating column (Cytiva, USA) charged with Ni$^{2+}$.

The column was washed with lysis buffer containing 0.05% DDM and 50 mM imidazole, followed by lysis buffer supplemented with 0.01% DDM. Peptidisc[45,46] (1 mL) at a concentration of 5 mg/mL in lysis buffer was injected onto the column and incubated at 4 °C for 1 h. The column was washed with 10 mL of lysis buffer containing 1 mg/mL peptide disc. Elution of the 6×His-tagged EmrAB-TolC protein complex was achieved with lysis buffer containing 300 mM imidazole. Buffer exchange of the eluate into Buffer-I, which consisted of 20 mM Tris (pH 7.5) and 150 mM NaCl, was conducted using a HiTrap Desalting column (Cytiva, USA). The 6×His-tagged EmrAB-TolC complex was further purified using ANTI-FLAG M2 affinity resin (GenScript, Nanjing, China). The resin was prepared by sequential washing with glycine HCl (pH 3.5) and Buffer-I. The protein solution and resin were mixed, gently rotated for 1 h at 4 °C, loaded into a chromatography column, and washed with Buffer-I. The mixture was resuspended in 1 mg/mL FLAG-peptide in Buffer-I, incubated for 30 min, centrifuged, and the supernatant was loaded onto a mini chromatography column to remove residual resin. This step was repeated three times. Fractions containing EmrAB-TolC were pooled, concentrated using a centrifugal filter unit (Merck, Germany) with a molecular weight cut-off of 100 kDa, flash-frozen in liquid nitrogen and stored at −80 °C.

The plasmids pET_BA and pRSF_AC were used to co-express and purify the EmrAB-TolC pump-FA containing the EmrB-EmrA fusion protein, TolC, and wild-type free EmrA using the same procedure employed for the pump-EA.

## Antibiotic sensitivity assay

Single colonies of *E. coli* C43(DE3) *ΔacrAB* harboring the plasmids pETDuet/pRSFDuet, pET_BA/pRSF_AC, pET_BA/pRSF_ΔAC, pETDuet-*emrAB*/pRSFDuet-*tolC-FLAG*, pETDuet-*emrAB$_{D29N}$*/pRSFDuet-*tolC-FLAG*, pETDuet-*emrAB$_{R109A}$*/pRSFDuet-*tolC-FLAG*, pETDuet-*emrA$_{L30Q/L303Q}$B*/pRSFDuet-*tolC-FLAG*, pETDuet-*emrA$_{I313A/V315A/V316A}$B*/pRSFDuet-*tolC-FLAG*, pETDuet-*emrAB$_{N62A}$*/pRSFDuet-*tolC-FLAG*, pETDuet-*emrAB$_{V147A}$*/pRSFDuet-*tolC-FLAG*, pETDuet-*emrAB$_{V148A}$*/pRSFDuet-*tolC-FLAG*, or pETDuet-*emrAB$_{I288A}$*/pRSFDuet-*tolC-FLAG* were grown in fresh LB supplemented with 50 μg/mL kanamycin and 100 μg/mL ampicillin at 37 °C with shaking at 220 rpm. When the optical density at 600 nm (OD$_{600}$) reached 0.5–0.6, expression was induced with 100 μM IPTG, the temperature was reduced to 22 °C, and the incubation was continued for 1 h.

The cells harboring the constructs were harvested by centrifugation, resuspended in fresh medium to an OD$_{600}$ of 0.5, and 10-fold serially diluted. Then 5 μL of each culture was plated on LB agar containing 60 μM IPTG in the presence of 1 μg/mL nalidixic acid, 5 μg/mL nitroxoline, or 1 μg/mL CCCP. As a control, 5 μL of all suspensions were plated on LB agar without antibiotics. Cell growth was monitored after culture at 37 °C overnight. The assay was performed three times independently.

## Drug-proton antiport assay

The following procedure was used to prepare everted membrane vesicles for drug-proton antiport assays of EmrB variants. The assay was conducted as described previously[41,47–49]. Inside-out (everted) membrane vesicles were prepared from C43 (DE3) ΔacrAB cells expressing the EmrB variants. Briefly, bacterial cells were cultured in 250 ml LB medium at 37 °C to an OD600 of 0.5, then induced for protein expression by adding 0.2 mM IPTG at 37 °C for 2 h to an OD600 of 1.5. The cells were harvested by centrifugation at 7500 × *g*

for 10 min, resuspended in 20 ml of membrane vesicle buffer A (200 mM Tris-HCl, pH 8.0, 2 mM EDTA, 30% sucrose) at room temperature, and incubated with 100 µg/ml lysozyme for 5–10 min. After adding 4 mM $MgCl_2$ and DNase I, the cell spheroplasts were pelleted at $7500 \times g$ for 10 minutes, then resuspended in 25 ml of ice-cold membrane vesicle buffer B (100 mM HEPES-KOH, pH 7.5, 100 mM KCl, 10 mM $MgCl_2$), supplemented with 2 mM DTT and 250 µL protease inhibitor (K1007; APExBIO). The spheroplasts were lysed by five passages through a homogenizer (ATS, AH-NANO) at 400–500 bar. The unbroken cells and debris were pelleted at $7500 \times g$ for 10 min, and the membranes in the supernatant were pelleted at $150,000 \times g$ for 40 min at 4 °C in a Ti70 rotor (Beckman Coulter). The membrane pellet was resuspended in membrane vesicle buffer B by gently squirting the buffer toward the pellet with a 200 µL tip. The vesicles were then centrifuged at 3500 g at 4 °C for 30 s to remove the aggregates.

To ensure equal quantities of everted membrane vesicles were used for each measurement, vesicles containing 200 µg total membrane proteins were added to 2 ml of pre-warmed (30 °C) buffer (20 mM Tris, 140 mM KCl, 10 mM $MgCl_2$, pH 6.0) and 1 µM 9-Amino-6-chloro-2-methoxyacridine (ACMA). Membranes were equilibrated for 4 minutes prior to fluorescence measurement with excitation at 409 nm and emission at 474 nm using an FluoroMax-4 spectrofluorometer (HORIBA Scientific). Before adding the substrates, 2 mM lactate was used and equilibrated for 3 min to energize the membrane, quenching the ACMA fluorescence. Fluorescence dequenching was observed upon adding 100–250 µM of nalidixic acid (Nal) or 2-chlorophenylhydrazine hydrochloride (CHH), likely due to the extrusion of $H^+$ by antiporters translocating the drugs into the vesicles. To dissipate the transmembrane $H^+$ gradient, 5 mM $NH_4Cl$ was added. The fluorescence changes induced by lactate and $NH_4Cl$ further confirmed that similar amounts of everted vesicles of comparable quality were used in each measurement.

6xHis-tag was added to the N-terminus of EmrA or C-terminus of EmrB of EmrAB-TolC pump, respectively. The same procedure was used to prepare everted membrane vesicles for Western blot assays of EmrA/EmrB variants (Figs. 3e; 4g).

**Electron microscopy data collection**
For the EmrAB-TolC pump-EA cryo-EM assays, 3.5 µL aliquots of purified protein in peptidisc (protein concentration = 0.7 mg/mL) were added to glow-discharged holey carbon grids (Quantifoil Au R1.2/1.3, 300 mesh; Quantifoil Micro Tools GmbH). Blotting was performed with filter paper for 3.5 s to remove excess sample, and a Vitrobot Mark IV instrument (Thermo Fisher Scientific) was used for rapid freezing in a liquid ethane slush. A Titan Krios electron microscope (Thermo Fisher Scientific) operating at 300 kV coupled with a SerialEM and a Gatan K3-Summit detector (Gatan, Inc.) operating in super-resolution counting mode[50] were used to automatically collect zero-energy-loss images of frozen and hydrated grids. Using a slit width of 20 eV, a GIF-Quantum energy filter (Gatan) was applied to exclude inelastically scattered electrons. Using a dose rate of ~15.15 electrons $Å^{-2}$ $s^{-1}$ (~18 electrons pixel$^{-1}$ $s^{-1}$) at an adjusted magnification of ×45,871.6 (yielding a pixel size of 1.09 Å at the sample level) and a total dose of ~60 electrons $Å^{-2}$ at the sample, 60 movie frames were recorded. The final dataset comprised 4334 movie stacks with defocus values between −1.0 and −2.5 µm.

For the EmrAB-TolC pump-FA, a holey carbon grid (Quantifoil Au R1.2/1.3, 300 mesh) was overlaid with a homemade graphene monolayer and cleaned with UV/ozone at room temperature for 10 min using a Gatan SOLARUS (950) Plasma Cleaning System (Gatan, Inc.), ensuring hydrophilicity of the graphene grid. Next, 3.5 µL aliquots of the purified protein sample from the peptidisc (0.1 mg/mL concentration) were applied to the grid and incubated for 30 s. Excess sample was removed by blotting with filter paper for 3.5 s, followed by rapid freezing in liquid ethane slush using a Vitrobot Mark IV instrument. Frozen-hydrated EmrAB-TolC particles were subjected to automatic data collection using a Titan Krios electron microscope at 300 kV with SerialEM and a Gatan K3-Summit direct electron detector running in super-resolution counting mode at an adjusted magnification of 47,169.9, equating to a measured physical pixel size of 1.06 Å and a dose rate of ~16 electrons $Å^{-2}$ $s^{-1}$ (~18 electrons pixel$^{-1}$ $s^{-1}$). Exposures lasting 3.533 s were split into 60 movie frames, resulting in an accumulated dose of ~56.6 electrons $Å^{-2}$ at the sample. Using a defocus range of −1.0 to −2.5 µm, 3306 movie stacks were collected.

**Image processing**
For the EmrAB-TolC pump-EA, the super-resolution movie frames were adjusted, including correction for gain reference and 2× binning, followed by motion correction using MOTIONCORR2[51]. Merging of the aligned movie frames into micrographs then allowed estimation of the contrast transfer function (CTF) using CTFFIND4[52]. RELION v3.1.3 software was used for subsequent image processing steps[53,54]. Templates for automatic particle picking were derived from (reference-free) two-dimensional classification of a manually selected particle subset. To minimize reference bias, these templates underwent low-pass filtering to 20 Å, enabling the automatic selection of 816,894 particles from all the micrographs. Two-dimensional classification of the images yielded 480,902 suitable particles.

Additional image processing steps were carried out using CryoSPARC[55]. Particles chosen via two-dimensional classification within RELION were imported into CryoSPARC and subjected to two further rounds of two-dimensional classification to discard any obviously discrepant particles, yielding 56,006 particles for classification by ab initio reconstruction using CryoSPARC. The parameters for this classification included two-class ab initio reconstruction with specific settings (initial alignment resolution of 25 Å; maximum alignment resolution of 6 Å; initial minibatch size of 150; final size of 600; and class similarity of 0)[56]. A total of 33,460 particles contributed to the resulting 3D volume, which was subjected to nonuniform refinement, generating an EmrAB-TolC reconstruction at a resolution of 3.14 Å. The resulting map quality for the EmrA and TolC sections was high, but the quality for the EmrB section was low. The AlphaFold2[57]-generated EmrB model was subsequently transformed into a map using e2pdb2mrc.py in EMAN2[58] and subsequently aligned to the EmrB map section within the EmrAB-TolC reconstruction. A focus mask was generated from the AlphaFold2-predicted model of EmrB and low-pass filtered to a 20-Å cut-off. CryoSPARC 3D classification without alignment was subsequently performed using this mask, yielding five well-defined classes from an initial set of ten. Aligning the 3D volumes and corresponding 19,180 particles from these classes using the Align 3D Maps program in CryoSPARC resulted in a homogeneous reconstruction of EmrAB-TolC at a resolution of 3.45 angstroms, with an improved map quality for the EmrB section. Local refinement was then carried out using a focused mask on the EmrAB portion, leading to a refined EmrAB-TolC reconstruction at 3.13 Å with improved map quality for the EmrAB sections (map-1 of pump-EA; and Supplementary Fig. 2c, d). Local refinement was also conducted using a focused mask on the TolC and α-helical hairpin domains of EmrA, leading to a refined reconstruction at 3.14 Å with better map quality for the EmrA-TolC sections (map-2 of pump-EA; and Supplementary Fig. 2c, d).

The similar image processing procedures were employed for EmrAB-TolC pump-FA, which generated a reconstruction at 3.58 Å resolution (map-3 of pump-FA; Supplementary Fig. 3c, d; Supplementary Table 2), as well as a reconstruction at 3.66 Å resolution (map-4 of pump-FA).

Resolution estimation was accomplished via CryoSPARC using independently refined half-reconstructions. The criterion for Fourier shell correlation was set at 0.143. The ResMap wrapper in CryoSPARC was used to calculate variation in local resolution (Supplementary Figs. 2; 3)[59].

## Protein structure prediction and modeling

AlphaFold3, AlphaFold2 and AF-Cluster were all employed for protein structure predictions. High-confidence predictions were selected based on their per-residue confidence score and corresponding predicted aligned error plot. Unless stated otherwise, the top rank of 5 predictions is shown and is visualized in ChimeraX Version 1.7. AlphaFold3 Beta was used to generate models of full-length EmrB (UniProt P0AEJ0)[37]. AlphaFold3-Multimer Beta was used to generate models of EmrAB, consisting of one molecule of full-length EmrB (UniProt P0AEJ0) with six molecules of full-length EmrA (UniProt P27303). AF-Cluster was used to predict multiple conformations of EmrB[38]. An MSA of EmrB (UniProt P0AEJ0) consisting of ~6000 sequences was first generated using ColabFold[60]. These sequences were then clustered into ~300 groups based on similarity using DBSCAN. A total of 16 clusters of sufficient size (at least 30 sequences) were then subsequently predicted using AlphaFold2. These conformations were visualized and analyzed using both ChimeraX and ChimeraUCSF.

## Model building and refinement

The two half-maps of cryo-EM map-1 and map-2 for EmrAB-TolC pump-EA were used to perform local map sharpening with DeepEMhancer[61], respectively. The resulting modified ~~full~~ map-1 was segmented into six EmrA protomers, and one EmrB protomer; the resulting modified map-2 was segmented into three TolC protomers. The EmrAB section segmented from modified map-1 and TolC section segmented from modified map-2 were merged into a full map of EmrAB-TolC pump-EA. A model of the β-barrel domain without β-CL, and the lipoyl domain, generated by AlphaFold3[37], were fitted to the map of individual EmrA protomers using Chimera[62]. The α-helical hairpin domain, C-terminal α-helix, and β-CL of the β-barrel domain were manually built using Coot[63]. Each EmrA protomer model was subsequently refined with Rosetta[64].

A homology model of EmrB in the outward-open state was generated by AlphaFold3[37]. Chimera was used to fit the N- and C-domains of this homology model to the map. The EmrB model was subsequently refined using Rosetta.

The structure of trimeric TolC, derived from the MacAB-TolC pump (PDB code: 5NIK), was docked into the TolC section of the modified map-2 using Chimera. To improve the local fit to the map, manual adjustments were made to the model using Coot.

Models of individual components were fitted into the cryo-EM map of EmrAB-TolC pump-EA using Chimera. Adjustments were made to the entire EmrAB-TolC pump-EA model to reduce the clashes using ISOLDE[65]. The model was then undergone real-space refinement against this modified map in the Phenix package[66].

The two half-maps of cryo-EM map-3 and map-4 for EmrAB-TolC pump-FA were used to perform local map sharpening with DeepEMhancer, respectively. The EmrAB section of modified map-3 and TolC section of modified map-4 were merged into a full map of EmrAB-TolC pump-FA. The refined model of EmrAB-TolC pump-EA was fitted into the cryo-EM map of pump-FA. The model section of the N-terminal α-helix of EmrA, generated by AlphaFold3, was fitted into the map of the individual EmrA protomer. Model sections without defined map were deleted. To improve the local fit to the map, manual adjustments were made to the model using Coot.

Any Ramachandran outliers were manually corrected in Coot, and stereochemistry was ensured using MolProbity (Supplementary Table 2)[67].

All maps and models EmrAB-TolC pump presented originate from the EmrAB-TolC pump-EA system, unless explicitly stated otherwise.

## Molecular dynamic simulations

MD simulations were based on the outward-open EmrB conformation observed in the cryo-EM structure of the EmrAB-TolC assembly. EmrB was inserted into mixed POPE/POPG (3:1) membranes and water and KCl at 150 mM concentration were added to the simulation system using the CHARMM-GUI webserver[68]. The initial simulation box had dimensions of 9.9 nm in x- and y-direction and 12.7 nm in z-direction. The system was energy-minimized and equilibrated in NVT and NpT ensembles using Gromacs2022[69] together with the CHARMM-GUI equilibration protocol for membrane proteins[68]. Production simulations were then carried out in the NpT ensemble, where the temperature was maintained at $T = 310\,K$ using the Nosé-Hoover thermostat[70], and the pressure was maintained semi-isotropically at 1 bar using the Parrinello-Rahman barostat[71]. The CHARMM36m force field[72] was used for the protein, lipids and ions together with the TIP3P model[73] for water molecules. For each protonation state of D29, we performed three independent production simulations of 1 μs length each for further analysis. The resulting MD trajectories were analysed using Gromacs utilities, MDAnalysis[74] and PENSA[35].

We used Autodock Vina[75] within the graphical interface AMDock[76] to dock the substrates to 10 EmrB conformations obtained from our molecular dynamics simulations. The top-scoring poses from each docking cycle were converted into densities for display using MDAnalysis[74].

## Reporting summary

Further information on research design is available in the Nature Portfolio Reporting Summary linked to this article.

## Data availability

Coordinates have been deposited in the Protein Data Bank (PDB) under PDB codes 8ZAL (EmrAB-TolC pump-EA) and 8ZAR (EmrAB-TolC pump-FA). Cryo-EM maps have been deposited in the Electron Microscopy Data Bank (EMD) under EMD codes EMD-39879 (EmrAB-TolC pump-EA) and EMD-39885 (EmrAB-TolC pump-FA). Molecular dynamics data are in the Figshare repository with the doi: 10.6084/m9.figshare.31240081 [doi.org/10.6084/m9.figshare.31240081]. Source data are provided with this paper.

## Code availability

Molecular Dynamics input files, coordinates and simulation trajectories can be accessed at doi: 10.6084/m9.figshare.31240081 [doi.org/10.6084/m9.figshare.31240081].

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

## Acknowledgements
This work was supported by the National Key R&D Program of China (2022YFC2303200); National Natural Science Foundation of China (31971133 to D.D.; 32270064 and 92478118 to Y.C.), the Science and Technology Commission of Shanghai Municipality (19PJ1407900, 19JC1414000 and 22WZ2504100 to D.D.; 24ZR1493200 to Y.C.), and the Chinese Academy of Sciences (XDB0570000 and 176002GJHZ2022022MI to Y.C.). BFL was supported by ERC Advanced grant (742210) and a Wellcome Trust Investigator award (200873/Z/16/Z). MLJ is supported by a UK Medical Research Council Intramural Programme Award MC_UU_000254/4 (RG94521).. Cryo-EM data were collected at the Bio-Electron Microscopy Facility of ShanghaiTech University with the assistance of Q. Sun, D. Liu, Z. Zhang, L. Wang and Y. Yang. We thank the Molecular Imaging Core Facility, the Molecular and Cell Biology Core Facility, and the Multi-Omics Core Facility at the School of Life Science and Technology for providing technical support. We are also grateful for the support of Lajos Kalmar of the MRC Toxicology Unit in the use of high-performance computing used in this study. We thank Sofiya Mason for help with molecular docking.

## Author contributions
Z.Z. performed cloning and overexpression of the EmrAB-TolC complex; Z.Z. and T.M. purified the EmrAB-TolC complex, prepared cryo-EM samples, collected cryo-EM data, determined structures, performed model building and refinement and prepared figures for the manuscript; J.G. and X.G. carried out drug-proton antiport assay; W.W., X.G., W.S., Q.W., J.G., S.L., H.L. and Q.O. optimized in-column peptide-disc methods, prepared homemade graphene monolayer grids, and assisted the collection of cryo-EM data; R.D., H.J., Z.Z., T.M., X.G., S.Z. and W.S. performed antibiotics sensitivity assays; T.M. and H.L. carried out Western blot assay; Y.C. supervised antibiotics sensitivity assays and discussed project design; M.L.J. performed protein structure predictions and modeling; U.Z. performed and analysed the molecular dynamics simulations; D.D. and B.L. conceived the project; D.D. designed and supervised all experiments; D.D., Y.C. and B.L. wrote the manuscript. All the authors contributed to the data interpretation and manuscript preparation.

## Competing interests
The authors declare no competing interests.
