## [Transparent Peer Review file · Nature Communications]

A model for drug transport across two membranes of Gram-negative bacteria by an MFS tripartite assembly

Corresponding Author: Professor Ben Luisi

Version 0:

Reviewer comments:

Reviewer #1

(Remarks to the Author)

The authors have addressed all my queries and the manuscript clarity has significantly improved.

Reviewer #2

(Remarks to the Author)

The authors did a great job addressing all of the comments. I am satisfied with the current version of the manuscript.

Just as a minor comment, I personally would still improve the depth of the figures - the main figures 1-3 could use some depth cue. To me these structure figures look somewhat flat. However, I would not insist on changing them, in case the authors themselves are happy with the look of their figures.

Reviewer #3

(Remarks to the Author)

The authors have substantially revised the manuscript and have addressed all of my previous comments and concerns. Although the revisions required a considerable amount of work, I feel the effort was worthwhile. The implemented molecular dynamics simulations nicely complement the experimental data. I recommend the manuscript for publication.

Reviewer #4

(Remarks to the Author)

This is an impressive and substantially improved revision. The structural, functional and computational analyses together provide a compelling and timely view of the EmrAB–ToIC MFS tripartite efflux pump and its role in multidrug resistance of Gram negative bacteria.

The authors have addressed the previous referee reports very thoroughly. In particular:

- The construct design is now clearly illustrated, and the new plate survival assays for wild type EmrAB–ToIC, pump EA and pump FA convincingly demonstrate that pump FA is functional and pump EA is not, clarifying the interpretation of the engineered constructs (Extended Data Fig. 1).
- Nomenclature and helix numbering have been standardised (TM1–TM12, TM A/B, periplasmic extended helix bundle, N and C domains), which greatly improves readability.
- The cryo EM analysis is now fully documented, with detailed workflows, angular distributions, local resolution estimates, map–model FSC curves and representative map–model overlays for key regions of EmrA and EmrB in both pump EA and pump FA (Extended Data Figs. 2–5).
- The interaction between the EmrA N terminal helix and EmrB has been analysed and visualised (Extended Data Fig. 18), and the structural comparison between pump EA and pump FA indicates that the EA modifications do not grossly distort the overall architecture.

- The new comparative section relating EmrB to QacA and MHA2168, and the systematic analysis of the EmrA β CL region and EmrB periplasmic helix bundle, place the authors' findings nicely in the broader context of DHA2 family transporters.
- The AlphaFold2/3 and AF Cluster results are now clearly separated from experimentally observed states, and the use of mutagenesis and docking to support the predicted inward open state is carefully explained.
- The additional MD simulations and SSI analysis around D29 convincingly show long range coupling between its protonation state and conformational changes in EmrB, including residues in the putative binding pocket.

In my view, the work is scientifically solid and suitable for publication in Nature Communications after a very minor textual revision. I list a few suggestions to further improve clarity and ensure the wording's strength is fully aligned with the available evidence.

1. Wording of the "one step mechanism"

Throughout the manuscript, the authors appropriately compare EmrAB–ToIC with RND-type and ABC-type tripartite pumps and highlight the architectural features that support a one-step transport model. At a few points, however, the language still reads as if a mechanistic pathway has been directly established. For example, in the Abstract, Introduction, and early Results, the authors describe that the architecture "uncovers a one-step mechanism that directly transports antimicrobial drugs across the entire envelope of Gram-negative bacteria."

I suggest softening such sentences slightly, for example, to formulations like:

- "supports a one step drug transport model across the entire envelope"
 - "is consistent with a one step transport pathway that bypasses the periplasm"
- while keeping the clearer "two step versus one step" comparison in the Discussion and Extended Data Fig. 22. This adjustment would better reflect that only the apo outward open state is structurally observed and that dynamic transport steps are inferred.

2. Clarifying the role of D29 considering the new vesicle assays

The new everted membrane vesicle experiments are an important addition and nicely demonstrate that the full EmrAB–ToIC complex mediates substrate-dependent proton flux, whereas EmrB alone or EmrAB without ToIC does not. At the same time, it is striking that the D29N/D221N double mutant retains proton transport activity comparable to wild type, whereas single point mutations at D29 or R109 largely abolish resistance in plate assays.

It would be helpful to explicitly discuss this apparent discrepancy in the main text. For example:

- make clear that D29 and R109 are clearly critical for overall pumping activity in the cellular context, but
- the vesicle data suggest that proton antiport can still occur when D29 and D221 are neutralized, implying that proton coupling may involve a more distributed network of residues.

On this basis, the authors might consider slightly tempering statements that present D29 as the proton carrier and instead refer to it as an important contributor within a broader proton coupling motif.

3. AlphaFold based inward open state

The use of AlphaFold3 and AF Cluster to predict an inward open state of EmrB is one of the most interesting aspects of the study, and the authors are careful to distinguish between observed and predicted states. I would nevertheless recommend a minor refinement of some phrases that could be read as a bit strong, for example:

- "AlphaFold3 had been able to predict capture the inward open state previously unobservable in the experimental data set."

It could be softened to something like:

- "AlphaFold3 predicts a plausible inward open state that has not yet been observed experimentally, and which is consistent with our mutagenesis data and with known MFS transport mechanisms."

A brief sentence in the Discussion explicitly stating that these AF3/AF Cluster conformations provide testable hypotheses for future structural and functional work would also be useful.

4. Allosteric phrasing

In the Results, the authors have removed explicit claims of allosteric regulation, which I agree with. In the Discussion, the authors still refer to the possibility that interactions between EmrA and EmrB allosterically modulate EmrB activity. This is a reasonable interpretation of the structural and SSI data. To maintain consistency with earlier changes, it might be worth reinforcing that this is a proposed allosteric communication pathway, supported by correlated motions and interface mapping but not yet directly tested in specific allosteric experiments.

5. Reproducibility of simulations and predictive models

Finally, given the central role of MD and AlphaFold predictions, it would be helpful to add a brief line to the Data availability section indicating whether MD trajectories, AF3 input files or docking ensembles are available from the corresponding authors, even if they are not deposited in a public repository (which I strongly recommend the authors do). Also, a short sentence or two describing the limitations of these methods. With these small clarifications, I believe the manuscript will be in excellent shape and will provide a valuable reference for both the efflux pump and MFS transporter communities.

We thank the reviewer for careful reading of the manuscript and the helpful comments. We have made further modifications to the manuscript in light of the recommendations by the reviewer.

Reviewer #4 (Remarks to the Author)

This is an impressive and substantially improved revision. The structural, functional and computational analyses together provide a compelling and timely view of the EmrAB–TolC MFS tripartite efflux pump and its role in multidrug resistance of Gram negative bacteria.

The authors have addressed the previous referee reports very thoroughly. In particular:

- The construct design is now clearly illustrated, and the new plate survival assays for wild type EmrAB–TolC, pump EA and pump FA convincingly demonstrate that pump FA is functional and pump EA is not, clarifying the interpretation of the engineered constructs (Extended Data Fig. 1).
- Nomenclature and helix numbering have been standardised (TM1–TM12, TM A/B, periplasmic extended helix bundle, N and C domains), which greatly improves readability.
- The cryo EM analysis is now fully documented, with detailed workflows, angular distributions, local resolution estimates, map–model FSC curves and representative map–model overlays for key regions of EmrA and EmrB in both pump EA and pump FA (Extended Data Figs. 2–5).
- The interaction between the EmrA N terminal helix and EmrB has been analysed and visualised (Extended Data Fig. 18), and the structural comparison between pump EA and pump FA indicates that the EA modifications do not grossly distort the overall architecture.
- The new comparative section relating EmrB to QacA and MHAS2168, and the systematic analysis of the EmrA β CL region and EmrB periplasmic helix bundle, place the authors findings nicely in the broader context of DHA2 family transporters.
- The AlphaFold2/3 and AF Cluster results are now clearly separated from experimentally observed states, and the use of mutagenesis and docking to support the predicted inward open state is carefully explained.
- The additional MD simulations and SSI analysis around D29 convincingly show long range coupling between its protonation state and conformational changes in EmrB, including residues in the putative binding pocket.

In my view, the work is scientifically solid and suitable for publication in Nature Communications after a very minor textual revision. I list a few suggestions to further improve clarity and ensure the wording's strength is fully aligned with the available evidence.

1.Wording of the “one step mechanism”

Throughout the manuscript, the authors appropriately compare EmrAB–TolC with RND-type and ABC-type tripartite pumps and highlight the architectural features that support

a one-step transport model. At a few points, however, the language still reads as if a mechanistic pathway has been directly established. For example, in the Abstract, Introduction, and early Results, the authors describe that the architecture “uncovers a one-step mechanism that directly transports antimicrobial drugs across the entire envelope of Gram-negative bacteria.”

I suggest softening such sentences slightly, for example, to formulations like:

- “supports a one step drug transport model across the entire envelope”
 - “is consistent with a one step transport pathway that bypasses the periplasm”
- while keeping the clearer “two step versus one step” comparison in the Discussion and Extended Data Fig. 22. This adjustment would better reflect that only the apo outward open state is structurally observed and that dynamic transport steps are inferred.

Our reply: We have modified the wording as suggested.

2. Clarifying the role of D29 considering the new vesicle assays

The new everted membrane vesicle experiments are an important addition and nicely demonstrate that the full EmrAB–TolC complex mediates substrate-dependent proton flux, whereas EmrB alone or EmrAB without TolC does not. At the same time, it is striking that the D29N/D221N double mutant retains proton transport activity comparable to wild type, whereas single point mutations at D29 or R109 largely abolish resistance in plate assays.

It would be helpful to explicitly discuss this apparent discrepancy in the main text. For example:

- make clear that D29 and R109 are clearly critical for overall pumping activity in the cellular context, but
- the vesicle data suggest that proton antiport can still occur when D29 and D221 are neutralized, implying that proton coupling may involve a more distributed network of residues.

On this basis, the authors might consider slightly tempering statements that present D29 as the proton carrier and instead refer to it as an important contributor within a broader proton coupling motif.

Our reply: We have changed the wording as recommended.

3. AlphaFold based inward open state

The use of AlphaFold3 and AF Cluster to predict an inward open state of EmrB is one of the most interesting aspects of the study, and the authors are careful to distinguish between observed and predicted states. I would nevertheless recommend a minor refinement of some phrases that could be read as a bit strong, for example:

- “AlphaFold3 had been able to predict capture the inward open state previously unobservable in the experimental data set.”

It could be softened to something like:

- “AlphaFold3 predicts a plausible inward open state that has not yet been observed experimentally, and which is consistent with our mutagenesis data and with known MFS transport mechanisms.”

Our reply:

We have made the changes as suggested.

A brief sentence in the Discussion explicitly stating that these AF3/AF Cluster conformations provide testable hypotheses for future structural and functional work would also be useful.

4. Allostery phrasing

In the Results, the authors have removed explicit claims of allosteric regulation, which I agree with. In the Discussion, the authors still refer to the possibility that interactions between EmrA and EmrB allosterically modulate EmrB activity. This is a reasonable interpretation of the structural and SSI data. To maintain consistency with earlier changes, it might be worth reinforcing that this is a proposed allosteric communication pathway, supported by correlated motions and interface mapping but not yet directly tested in specific allosteric experiments.

Our reply: we have changed the wording.

5. Reproducibility of simulations and predictive models

Finally, given the central role of MD and AlphaFold predictions, it would be helpful to add a brief line to the Data availability section indicating whether MD trajectories, AF3 input files or docking ensembles are available from the corresponding authors, even if they are not deposited in a public repository (which I strongly recommend the authors do). Also, a short sentence or two describing the limitations of these methods. With these small clarifications, I believe the manuscript will be in excellent shape and will provide a valuable reference for both the efflux pump and MFS transporter communities.

Our reply: the data availability has been included.